

# An Extensible Perturbed Parameter Ensemble (PPE) for the Community Atmosphere Model Version 6

Trude Eidhammer[1], Andrew Gettelman[1,2], Katherine Thayer-Calder[1], Duncan Watson-Parris[3],
Gregory Elsaesser[4,5], Hugh Morrison[1], Marcus van Lier-Walqui[4,5], Ci Song[6], and Daniel McCoy[6]

[1]National Center for Atmospheric Research, Boulder, CO, USA
[2]Now at: Pacific Northwest National Laboratory, Richland, WA, USA
[3]Scripps Institution of Oceanography and Halıcıoğlu Data Science Institute, University of California San Diego, La Jolla, CA,
USA
[4]Columbia University, New York, NY, USA
[5]NASA Goddard Institute for Space Studies, New York, NY, USA
[6]Department of Atmospheric Science, University of Wyoming, Laramie, WY, USA

**Correspondence:** Trude Eidhammer (trude@ucar.edu)

**Abstract.** This paper documents the methodology and preliminary results from a Perturbed Parameter Ensemble (PPE) technique, where multiple parameters are varied simultaneously and the parameter values are determined with Latin hypercube sampling. This is done with the Community Atmosphere Model version 6 (CAM6), the atmospheric component of the Community Earth System Model version 2 (CESM2). We apply the PPE method to CESM2-CAM6 to understand climate sensitivity to atmospheric physics parameters. The initial simulations vary 45 parameters in the microphysics, convection, turbulence and aerosol schemes with 263 ensemble members. These atmospheric parameters are typically the most uncertain in many climate models. Control simulations are analyzed and targeted simulations to understand climate forcing due to aerosols and fast climate feedbacks. The use of various emulators is explored in the multi-dimensional space mapping input parameters to output metrics. Parameter impacts on various model outputs, such as radiation, cloud and aerosol properties are evaluated. Machine learning is also used to probe optimal parameter values against observations. Our findings show that using PPE is a valuable tool for climate uncertainty analysis. Furthermore, by varying many parameters simultaneously, we find that many different combinations of parameter values can produce results consistent with observations, and thus careful analysis of tuning is important. The CESM2-CAM6 PPE is publicly available, and extensible to other configurations to address questions of other model processes in the atmosphere and other model components (e.g. coupling to the land surface).

## 1 Introduction

General circulation models (GCMs) have numerous and long-standing biases due in part to uncertain representations of the physical processes (e.g., Trenberth and Fasullo, 2010). This is especially true for processes that occur at subgrid scales, such as microphysics, turbulence, convection and aerosol processes. Because these processes are not resolved, their effects on the grid-scale model state variables are represented via parameterizations, rather than explicitly solving the process equations at the natural scale of the phenomena being represented. For example, the evolution of a single cloud drop in a turbulent flow over





a small domain can be simulated explicitly, but the evolution of a cloud drop population cannot be directly simulated directly due to the sheer number of drops within each grid volume of typical atmospheric models (with grid spacing of 10's of m to 10's of km). Moreover, for many processes, including cloud and aerosol microphysics, even at the natural scale of the phenomenon (e.g., scale of an individual drop) there are uncertainties in the underlying physical processes. That is, for many processes there

are no governing equations at any scale. For example, for cloud microphysics there are fundamental uncertainties in how drops collide and either bounce, coalesce, or breakup. Most ice microphysical processes, including nucleation, diffusional growth, riming, and aggregation remain highly uncertain even at the scale of individual particles (e.g., Morrison et al., 2020).

Parameterizations typically include parameters whose values are constrained by theory, high-resolution process models, and/or observations. To varying degrees, these parameter values are uncertain because of both uncertainty in how to best rep-

resent the impact of subgrid-scale processes at the grid scale as well as fundamental uncertainty at the process scale. In climate models, parameter values are adjusted within the bounds of uncertainty to produce realistic output relative to observations. However, this process, usually referred to as "tuning", faces several challenges (Hourdin et al., 2016). For example, since climate GCMs comprise several different physics packages, finding the best parameter values in one physics package could impact the others and produce out of balance results. As a consequence, there may be a dependence on the sequence in which

the physics packages are tuned. As part of this process, it is important to understand how uncertainty in parameter values translates to uncertainty in simulated climate. Some parameters are more uncertain than others, but may have a relatively small or large impact on simulated climate across the range of this uncertainty.

Tuning, and the associated investigation of parameter uncertainties, can be done in several different ways. Each method has an associated computational cost, which is usually a consequence of how many simulations are performed. Traditionally,

sensitivity to parameters is analyzed using a "One At a Time" (OAT) method (Schmidt et al., 2017). When performed as part of model tuning process, this can represent an optimized random walk approaching the minimization of an informal cost function (errors against a sum of observations). OAT methods do not account for nonlinear relationships between different parameters and resulting outputs are generally inefficient. Furthermore, to perform simulations over the entire parameter space with many variable parameters, a large number of simulations are required. For example, in the current study we perturb 45 different

parameters, which would require a minimum of $3.5 \cdot 10^{13}$ ($2^{45}$) simulations using OAT if each parameter was tested with only two values in all combinations. The number of simulations needed increases exponentially if each parameter were perturbed with additional values, i.e., the number of required simulations is $M^N$ for OAT where $N$ is the number of parameters and $M$ is the number of values tested for each parameter.

Over the last several years, more objective and efficient methods have been developed to perturb multiple moist physics and

aerosol parameters simultaneously (Lee et al., 2011; Qian et al., 2015). These methods have been used to optimize models in an automated way (Jackson et al., 2008; Wagman and Jackson, 2018; Regayre et al., 2018; Peatier et al., 2022) and to understand model uncertainty (Posselt and Vukicevic, 2010; van Lier-Walqui et al., 2012; Regayre et al., 2014; Qian et al., 2015; Lee et al., 2016; Qian et al., 2018; Watson-Parris et al., 2020; Duffy et al., 2024). They provide a more robust platform for uncertainty quantification and objective improvement of climate models, ranging from parameter tuning to understanding

structural deficiencies of models, for example, when no combination of parameters converges to observations. Comprehensive





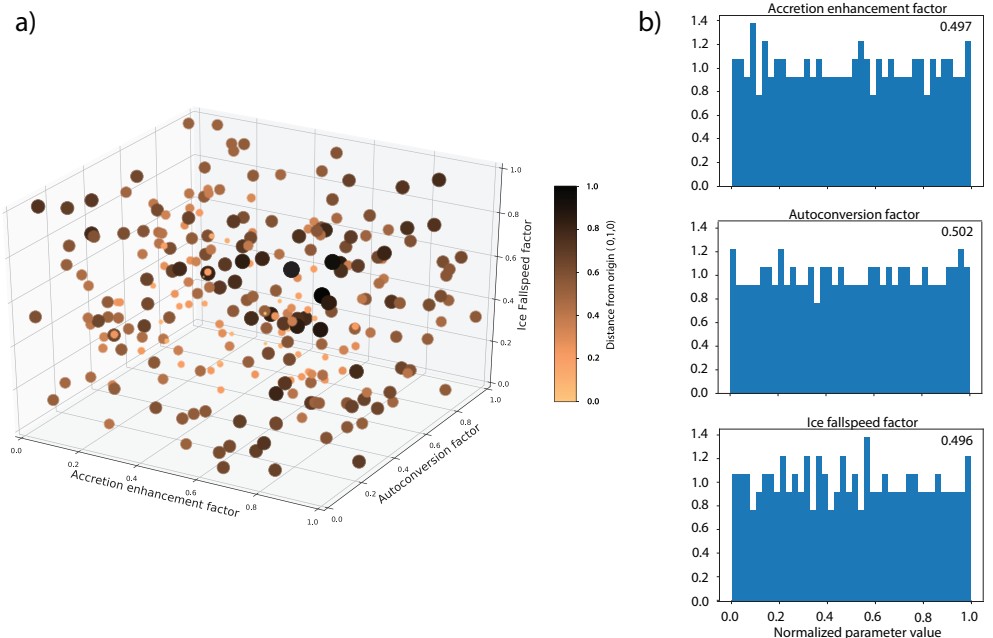

**Figure 1.** Example of Latin hypercube sampling with 3 normalized perturbed parameters (accretion enhancement factor, autoconversion factor and ice fallspeed factor). The color and size of the symbols in a) represent the Euclidean distance from the origin (0,1,0) for all 263 parameter sets in the full ensemble. Darker and larger symbols are located closer in the viewpoint. b) Normalized histogram of the marginals with the mean value over all parameter values shown in the upper right of each plot.

sets of perturbed parameter values can be used for development of sophisticated fast-model emulators to help with model tuning or even to advance process level understanding and guide selection of key additional data for constraining models (Regayre et al., 2018).

The goal of this manuscript is to document the methodology for creating a large perturbed parameter ensemble (PPE) with

the Community Earth System Model version 2 (CESM2; Danabasoglu et al., 2020) atmospheric component, the Community Atmosphere Model version 6 (CAM6; Gettelman et al., 2019). We will also present early results on PPE spread for certain outputs, key parameter sensitivities of the model and preliminary results of model emulation. The data, methods/scripts and code for reproducing and extending the PPE are now available to the community. Section 2 contains a description of the methodology used to create the PPE, including parameters and methods. Section 3 describes the method used for the modeling

and the emulators. Section 4 describes key initial results of the PPE, emulators and simple tuning, and section 5 provides a summary and conclusions.





## 2  PPE description

To generate the PPE, we created a set of variable parameters using Latin hypercube sampling (McKay et al., 2000). With this technique, random values are created within a determined range. Ranges of the possible values are divided into a number
of bins equal to the number of samples. Each parameter is assigned a value within a random bin, and no parameters from subsequent samples can have a value from previously sampled bins. In this way, the parameter sets for all samples cover the entire parameter range for each parameter and have marginal distributions that are uniformly distributed. Figure 1a shows an example of how three different parameters are sampled in relation to each other. The color and size of the symbols represent the distance from the center of origin in the plot to illustrate the depth of the plot. Note that the points in Figure 1a are generally
uniformly distributed in the 3-D space, and have uniform marginal distributions in each dimension (Figure 1b), which is a key aspect of Latin hypercube sampling.

Using this sampling, we initially created 250 different sets of parameter values in addition to the default CESM2-CAM6 setup (total of 251 sets). After preliminary analysis of the initial simulations we decided to extend the range for one of the parameters ($micro\_mg\_max\_nicons$). The method we employed is general for any parameters with Latin hypercube sampling.
A relative Euclidean distance metric ($d$) was created. For each individual ensemble $j$, we calculate the average distance of each parameter $i$ in ensemble $j$ to parameter $i$ in the other ensembles. Then $d$ is the sum of all Euclidean distances in ensemble $j$ divided by number of parameters ($pa$) and ensembles ($en$):

$$d = \frac{\sum_{i=1}^{pa} \sum_{j=1, j \neq m}^{en} (p(i,m)^2 - p(i,j)^2)}{en \cdot pa} \qquad (1)$$

The relative Euclidean distance for the original 251 ensemble members are shown in Figure 2 (250 perturbation cases plus 1
default case).

We then generated 7500 new parameter sets. Out of these 7500 sets, we picked 12 sets where the single parameter value of $micro\_mg\_max\_nicons$ was within the new range and had the largest relative Euclidean distance value (equal to or greater than the average Euclidean distance between each of the original 251 parameter samples) to make a total of 263 PPE sets. The reason for choosing the sets with the largest relative Euclidean distance is to avoid the problem of close-proximity points.
The relative Euclidean distance of all the 7500 sets that had a parameter value within the new range and a relative Euclidean distance greater than the average distance (0.16) were met for only 32 of the 7500 sets.

Using this approach, we then archived the 262 parameter sets plus the default case in a single file with metadata. Every parameter was chosen to be run-time configurable (not hard-wired in code). A script for running CESM2-CAM6 was developed which sets up a model simulation, then copies ("clones") the configuration to a new name, and substitutes a parameter set from
the file. This method enables reproduction and extension of the PPE from a single file and script. CESM2 and CAM6 can be run in many different configurations (standard Atmosphere-Ocean for CAM6, fully coupled CESM2, aquaplanet, single column, nudged mode, etc). Archiving the parameter sets and the automated run script allows any CESM configuration to be run with the same parameter sets for different types of analysis or different diagnostic output.




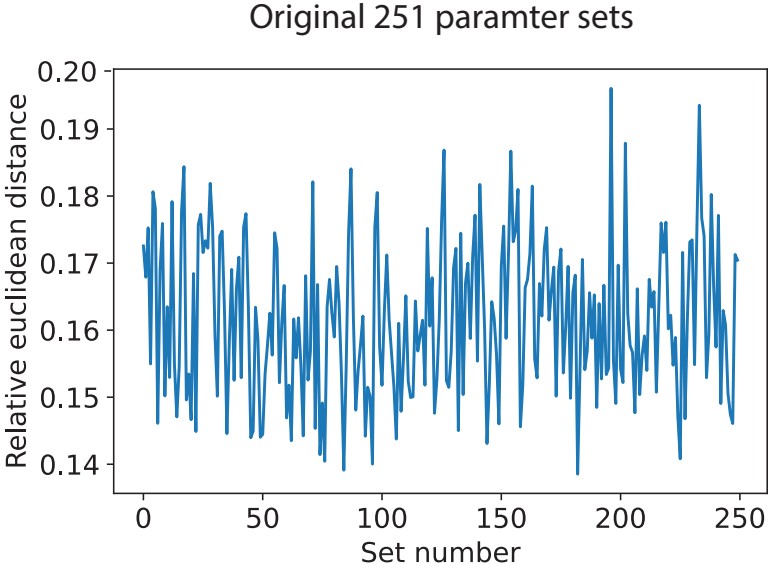

**Figure 2.** The relative Euclidean distance ($d$) for each of the original 251 parameter sets.

## 3 Methodology

Here we first describe the CESM2-CAM6 model, then the simulations conducted, parameters varied, and finally the emulators used on the model output.

### 3.1 Model description

In this study, we use the Community Earth System Model version 2 (CESM2; Danabasoglu et al., 2020), which contains the Community Atmosphere Model version 6 (CAM6). CAM6 uses a 4 mode version of the Liu et al. (2012) Modal Aerosol Model

(MAM4) with modifications to include stratospheric sulfur (Mills et al., 2016). This version has an extra mode for primary carbon, and has a better representation of black carbon and sulfate evolution. Cloud microphysics in CAM6 uses version 2 of the Morrison and Gettelman (2008) scheme, described by Gettelman and Morrison (2015) and Gettelman et al. (2015). CAM6 replaces the CAM5 shallow convection, planetary boundary layer and cloud macrophysics schemes with a new unified turbulence scheme, the Cloud Layers Unified By Binormals (CLUBB), originally developed by Golaz et al. (2002) and integrated

in CAM by Bogenschutz et al. (2013). CAM6 also features a new mixed phase ice nucleation scheme developed by Hoose et al. (2010). Deep convection is represented by the Zhang and McFarlane (1995) scheme. These CAM6 parameterizations have been implemented in CESM2 as described in Bogenschutz et al. (2018).





## 3.2 Simulations

We conducted three sets of simulation ensembles using the parameter samples. The first set uses near present day cyclic
boundary conditions for the year 2000. The greenhouse gases and atmospheric oxidants are average values for the 1995-2005
period. The average monthly sea surface temperatures (SSTs) for 1995-2010 are used. Emission of aerosols and precursors is
also set to 1995-2005 in the present day (PD) simulation. The second set of simulations is the pre-industrial (PI) configuration.
This uses the same setup as PD, but the aerosol emission is estimated for the year 1850. In the third set of simulations the
PD configuration is used again, but the SST is uniformly increased by 4K (SST4K). All simulations use a resolution of 0.9°
latitude × 1.1° longitude with 32 levels in the vertical up to 10 hPa. By performing these three sets of simulations with the
same parameter sets, not only can we evaluate the output spread by perturbing parameters, but we can also evaluate the aerosol
forcing (difference between PD and PI) and cloud feedback (difference between PD and SST4). We tested two different run
lengths (3 and 5) and found that we could reproduce (emulate) a given 2-D field with similar RMSE using 3 or 5 years of
simulation. All simulations presented herein are 3 years long.

Model output is archived monthly and daily for select fields. Output is available at: https://doi.org/10.26024/bzne-yf09
(Eidhammer et al., 2022). Also available is a python script to create the parameter file and scripts to submit the PPE simulations.

## 3.3 Parameters

All three simulation sets are run with 263 different ensemble members corresponding to the sets of 45 perturbed parameters
plus the default parameter set as described in section 2. Ensemble member 0 is the standard (default) CESM2-CAM6 setup. The
remaining 262 ensemble members are run with parameters determined with the Latin hypercube sampling where the minimum
and maximum values are given in Table 1. The values in Table 1 are the physical values, while the Latin hypercube sampling
use the normalized ranges to determine the parameter values. The range of values in Table 1 are chosen by "expert elicitation"
among the parameterization developers for cloud microphysics, convection, unified turbulence and aerosol activation.

The parameters encompass most of the moist physical parameterizations and aerosols. This includes the unified turbulence
closure (CLUBB; Golaz et al., 2002), the cloud microphysics (MG2; Gettelman and Morrison, 2015), the Modal Aerosol
Model (MAM; Liu et al., 2012) and the Zhang-McFarlane deep convection scheme (ZM; Zhang and McFarlane, 1995).

A brief description of the CLUBB parameters is found in Guo et al. (2014). $clubb\_C2rt$ is the damping of scalar variances
for liquid water; increasing this makes CLUBB behave closer to complete or no cloudiness (no variance) and brightens clouds.
The parameters $clubb\_C6rt$ and $clubb\_C6thl$ describe damping of scalar water fluxes for smaller liquid water path than the
mean. Both parameters were perturbed simultaneously so that $clubb\_C6rt=clubb\_C6thl$. Decreasing these parameters tends
to boost fluxes, producing a more well mixed layer, with minor effects on cloud brightness. $clubb\_C6rtb$ and $clubb\_C6htlb$ are
the damping of scalar water fluxes for larger liquid water path then the mean, with similar effects as the damping parameters
for small liquid water path. Again, both parameters were perturbed with the same values. $clubb\_C8$ describes the dissipation
of skewness of the vertical velocity; increasing this parameter reduces skewness, which brightens clouds. $clubb\_beta$ sets the
plume widths for liquid water potential temperature and total water. An increase in $clubb\_beta$ leads to an increase in the scalar



**Table 1.** A description of the parameters that are perturbed and their ranges.

| Physics Scheme | Parameter Name | Description | Default | Min | Max | Units |
|---|---|---|---|---|---|---|
| $CLUBB$ | clubb_C2rt | Damping on scalar variances | 1.0 | 0.2 | 2 | - |
| | clubb_C6rt | Low skewness in C6rt skewness function | 4.0 | 2.0 | 6 | - |
| | clubb_C6rtb | High skewness in C6rt skewness function | 6.0 | 2.0 | 8 | - |
| | clubb_C6thl | Low skewness in C6thl skewness function | 4.0 | 2.0 | 6 | - |
| | clubb_C6thlb | High skewness in C6thl skewness function | 6.0 | 2.0 | 8 | - |
| | clubb_C8 | Coef. #1 in C8 skewness Equation | 4.2 | 1.0 | 5 | - |
| | clubb_beta | Set plume widths for theta_l and rt | 2.4 | 1.6 | 2.5 | - |
| | clubb_c1 | Low Skewness in C1 Skw. | 1.0 | 0.4 | 3 | - |
| | clubb_c11 | Low Skewness in C11 Skw | 0.7 | 0.2 | 0.8 | - |
| | clubb_c14 | Constant for u'$^2$ and v'$^2$ terms | 2.2 | 0.4 | 3 | - |
| | clubb_c_K10 | Momentum coefficient of Kh_zm | 0.5 | 0.2 | 0.6 | - |
| | clubb_gamma_coef | Low Skw.: gamma coef. Skw | 0.308 | 0.25 | 0.35 | - |
| | clubb_wpxp_L_thresh | Lscale threshold, damp C6 and C7 | 60 | 20 | 200 | m |
| $MG2$ | micro_mg_accre_enhan_fact | Accretion enhancing factor | 1.0 | 0.1 | 10.0 | - |
| | micro_mg_autocon_fact | Autoconversion factor | 0.01 | 0.005 | 0.2 | - |
| | micro_mg_autocon_lwp_exp | KK2000 LWP exponent | 2.47 | 2.10 | 3.30 | - |
| | micro_mg_autocon_nd_exp | KK2000 autoconversion exponent | -1.1 | -0.8 | -2 | - |
| | micro_mg_berg_eff_factor | Bergeron efficiency factor | 1.0 | 0.1 | 1.0 | - |
| | micro_mg_dcs | Autoconversion size threshold ice-snow | 500e-06 | 50e-06 | 1.000e-06 | m |
| | micro_mg_effi_factor | Scale effective radius for optics calculation | 1.0 | 0.1 | 2.0 | - |
| | micro_mg_homog_size | Homogeneous freezing ice particle size | 25e-6 | 10e-6 | 200e-6 | m |
| | micro_mg_iaccr_factor | Scaling ice/snow accretion | 1.0 | 0.2 | 1.0 | - |
| | micro_mg_max_nicons | Maximum allowed ice number concentration | 100e6 | 1e5 | 10000e6 | # kg$^{-1}$ |
| | micro_mg_vtrmi_factor | Ice fall speed scaling | 1.0 | 0.2 | 5.0 | m s$^{-1}$ |
| $Aerosol$ | microp_aero_npccn_scale | Scale activated liquid number | 1 | 0.33 | 3 | - |
| | microp_aero_wsub_min | Min subgrid velocity for liq activation | 0.2 | 0 | 0.5 | m s$^{-1}$ |
| | microp_aero_wsub_scale | Subgrid velocity for liquid activation scaling | 1 | 0.1 | 5 | - |
| | microp_aero_wsubi_min | Min subgrid velocity for ice activation | 0.001 | 0 | 0.2 | m s$^{-1}$ |
| | microp_aero_wsubi_scale | Subgrid velocity for ice activation scaling | 1 | 0.1 | 5 | - |
| | dust_emis_fact | Dust emission scaling factor | 0.7 | 0.1 | 1.0 | - |
| | seasalt_emis_scale | Seasalt emission scaling factor | 1.0 | 0.5 | 2.5 | - |
| | sol_factb_interstitial | Below cloud scavenging of interstitial modal aerosols | 0.1 | 0.1 | 1 | - |
| | sol_factic_interstitial | In-cloud scavenging of interstitial modal aerosols | 0.4 | 0.1 | 1 | - |
| $ZM$ | cldfrc_dp1 | Parameter for deep convection cloud fraction | 0.1 | 0.05 | 0.25 | - |
| | cldfrc_dp2 | Parameter for deep convection cloud fraction | 500 | 100 | 1.000 | - |
| | zmconv_c0_lnd | Convective autoconversion over land | 0.0075 | 0.002 | 0.1 | m$^{-1}$ |
| | zmconv_c0_ocn | Convective autoconversion over ocean | 0.03 | 0.02 | 0.1 | m$^{-1}$ |
| | zmconv_capelmt | Triggering threshold for ZM convection | 70 | 35 | 350 | J kg$^{-1}$ |
| | zmconv_dmpdz | Entrainment parameter | -1.0e-3 | -2.0e-3 | -2.0e-4 | m$^{-1}$ |
| | zmconv_ke | Convective evaporation efficiency | 5.0e-6 | 1.0e-6 | 1.0e-5 | (kg m$^{-2}$ s$^{-1}$)$^{0.5}$ s$^{-1}$ |
| | zmconv_ke_lnd | Convective evaporation efficiency over land | 1.0e-5 | 1.0e-6 | 1.0e-5 | (kg m$^{-2}$ s$^{-1}$)$^{0.5}$ s$^{-1}$ |
| | zmconv_momcd | Efficiency of pressure term in ZM downdraft CMT | 0.7 | 0 | 1 | - |
| | mconv_momcu | Efficiency of pressure term in ZM updraft CMT | 0.7 | 0 | 1 | - |
| | zmconv_num_cin | Allowed number of negative buoyancy crossings | 1 | 1 | 5 | - |
| | zmconv_tiedke_add | Convective parcel temperature perturbation | 0.5 | 0 | 2 | K |



skewness. This affects liquid water and cloud fraction. $clubb\_c1$ is the skewness of the lower side of the C1 skewness function (standard deviation of vertical velocity); increasing $clubb\_c1$ dims clouds. $clubb\_c11$ is the low skewness for buoyancy damping of vertical velocity. Increasing $clubb\_c11$ brightens clouds. $clubb\_c14$ is a constant for dissipation of $u'^2$ and $v'^2$ (variances of the horizontal velocity components), and lower values brighten clouds. $clubb\_c\_K10$ is a coefficient in the momentum

equation. An increase in $clubb\_c\_K10$ increases the eddy diffusivity of momentum, which, in turn, increases near-surface wind magnitude. $clubb\_gamma\_coef$ controls the skewness of the vertical velocity distributions (different moments), and lowering it brightens low clouds. $clubb\_wpxp\_L\_thresh$ is a threshold for turbulent mixing length, below which extra damping is applied to scalar fluxes. A higher value means that the extra damping is applied to a greater range of mixing lengths.

The MG2 microphysics scheme (Gettelman and Morrison, 2015; Gettelman et al., 2015) takes bulk water and divides it into

four hydrometeor categories (cloud liquid, ice, rain and snow), predicting mass and number mixing ratios for each. Several parameters are used to control the rain formation processes of autoconversion and accretion. Autoconversion is the coalescence of cloud droplets that become rain and it is dependent on the cloud water mass mixing ratio ($q_d$) and inversely dependent on drop number ($N_d$). $micro\_mg\_autocon\_lwp\_exp$ alters the exponent on $q_d$ and $micro\_mg\_autocon\_nd\_exp$ alters the exponent on $N_d$. $micro\_mg\_autocon\_fact$ linearly scales autoconversion. Accretion is the process of rain drops collect-

ing cloud water. $micro\_mg\_accre\_enhan\_fact$ linearly scales it. $micro\_mg\_berg\_eff\_factor$ scales the rate of vapor deposition onto ice (which also impacts supercooled liquid). $micro\_mg\_max\_nicons$ is the maximum allowed ice number concentration. $micro\_mg\_dcs$ is the threshold diameter for cloud ice to autoconvert to snow. $micro\_mg\_iaccr\_factor$ similarly scales the accretion of cloud ice by snow. $micro\_mg\_effi\_factor$ scales the size used for the optics calculation for cloud ice. $micro\_mg\_homog\_size$ alters the initial size generated when liquid homogeneously freezes to ice. Finally

$micro\_mg\_vtrmi\_factor$ linearly scales the ice and snow fall speed.

Several parameters are related to aerosols, mostly aerosol emissions, cloud particle nucleation, and scavenging. $microp\_aero\_npccn\_scale$ scales the activated cloud condensation nuclei (CCN) concentration, affecting drop number concentration. Sub-grid scale vertical velocities are used for both cloud droplet activation ($wsub$) and ice nucleation ($wsubi$), and are derived from the turbulent kinetic energy (TKE) calculation in CLUBB. This calculation applies maximum and minimum limits to the sub-grid vertical

velocities. Here, we perturb the minimum values which are set with $microp\_aero\_wsub\_min$ and $microp\_aero\_wsubi\_min$. The sub-grid vertical velocities are linearly scaled with $microp\_aero\_wsub\_scale$ and $microp\_aero\_wsubi\_scale$. Higher sub-grid vertical velocities will generally activate more aerosol leading to higher drop and crystal numbers. Dust emissions are linearly scaled with $dust\_emis\_fact$ and sea-salt emissions scaled with $seasalt\_emis\_scale$. Finally, the scavenging of aerosols in clear air below cloud by precipitation is scaled by $sol\_factb\_interstitial$ and within cloud by $sol\_factic\_interstitial$.

Deep moist convection is parameterized by Zhang and McFarlane (1995), referred to as ZM. $cldfrc\_dp1$ and $cldfrc\_dp2$ define the shape of the relationship between convective mass flux and convective cloud fraction ($dp1$=linear term, $dp2$=log term). An increase in either of these parameters increases convective cloud fraction. Autoconversion of convective condensate to precipitation increases by increasing $zmconv\_c0\_lnd$ (over land) and $zmconv\_c0\_ocn$ (over ocean), increasing the efficiency of convective precipitation. $zmconv\_capelmt$ is the Convective Available Potential Energy (CAPE) triggering threshold for deep

convection, where a higher value triggers less often and allows more CAPE to build up. $zmconv\_dmpdz$ changes the entrain-



ment rate for the initial parcel buoyancy test, and a larger value means more mixing and damped convection. $zmconv\_ke$ is the convective evaporation efficiency over ocean and $zmconv\_ke\_lnd$ over land. Larger values mean more evaporation. There are two parameters for the pressure term in the convective momentum transport equation; $zmconv\_momcd$ is for downdrafts and $zmconv\_momcu$ for updrafts. Increasing them reduces the impact of resolved vertical wind shear. $zmconv\_num\_cin$ is

the allowed number of negative buoyancy crossings before the convective top is reached. Larger values mean deeper convection. Finally, $zmconv\_tiedke\_add$ is a convective parcel temperature perturbation, where a higher value means more buoyant parcels and deeper convection.

## 3.4    Emulator description

We perform analysis of the raw model output across the PPE and also use several different emulation tools to analyze the

ensemble. This is done to show potential use of machine learning on the PPE. Here we focus on how well fast emulators can reproduce specific model features, and then show a simple demonstration of how they can be used to tune the model. We utilize two separate emulator toolkits. The first is the Earth System Emulator (ESEm), which is an open source tool providing a general workflow for emulating and validating a wide variety of models and outputs (Watson-Parris et al., 2021). This tool uses well-established libraries for the emulation of general circulation models with different regression techniques (neural network,

Gaussian process, and random forest) and provides hardware optimised functions for efficiently sampling them. The tool also features the ability to train on 2 dimensional (2-D) fields of data. The second is a neural network emulator (hereafter referred to as the Columbia NN or Columbia emulator) developed for tuning the NASA GISS GCM (ModelE), with the final model being a combination of up to 12 different neural network models (or setups).

In our emulations, we used 210 simulations for training data (80%) and 52 simulations for test data (20%) with no separate

samples withheld for testing. Below are longer descriptions of the emulator techniques used here.

### 3.4.1    Neural network

Inspired by the human brain, neural networks (NN) form a class of flexible and expressive non-linear functions parameterised by a large number of weights. They generally consist of multiple layers of nodes connected by edges. Each node consists of a simple (differentiable) activation function which transforms weighed input into outputs for the following nodes. The weights

are optimised using gradient descent against the provided training data. The structure (or architecture) of the NN, including the number and connectivity of the layers, provides a strong inductive bias on the skill of the trained network.

The Columbia approach uses an ensemble of several NNs whose outputs are averaged. Tests showed that this methodology reduced emulator predictive noise and bias, relative to GCM output. The ensemble members are selected on the basis of minimum validation (mean square and mean absolute) error. Each NN uses a fully connected design whereby each node in

each layer is connected to every node in the next layer, sometimes referred to as a multi-layer perceptron (MLP). The activation function is either a Rectified Linear Unit (ReLU) or leaky-ReLU, depending on the NN used. The hyperparameters of each NN were chosen by manual iteration through various values of nodes-per-layer and choice of activation function (see below). The Adam optimizer was used with mean square error during training with a learning rate of 0.001. Early stopping with a





**Figure 3.** The ensemble zonal annual mean and ±1 standard deviation across the ensemble for A) aerosol optical depth (AOD), B) column ice water path (IWP), C) vertically integrated accumulation mode sulfate mass (BURDEN SO4), D) vertically integrated cloud condensation nuclei at 0.1% supersaturation (CCN 0.1%), E) column liquid water path (LWP), F) longwave cloud forcing (LWCF), G) cloud top number concentration for liquid (ACTNL) and H) a histogram of the global, annual global mean net top of atmosphere (TOA) flux balance across the 263 ensemble members. In A-G), solid lines show the ensemble means and ±1 standard deviation is indicated by the shading. Colored dotted lines (vertical in the TOA histograms) are the default cases. Orange: Present Day (PD), Blue: SST4K, Green: Pre-Industrial(PI).





patience of 100 epochs was used to prevent overfitting, using validation loss – typically, training required between 200 and
500 epochs. Most of these design choices were determined ad hoc. The Columbia emulator was originally designed to emulate
the effect of GCM parameter perturbations (45 for GISS ModelE) on the values of climatological GCM output performance
scores (36 scalar diagnostics for ModelE), with skill quantified using the equivalent satellite climatologies. This emulator was
not designed to output spatially resolved fields.

The ESEm NN tries to capture the spatial covariance of the full model output fields using a fully convolutional neural
network (CNN). Rather than being fully connected, which would lead to prohibitively many parameters, CNNs convolve small
kernels over the image to learn relevant features. While still requiring more parameters than assuming grid-point independence,
and hence more training data, we have found that with suitable normalisation such emulators can skillfully reproduce CAM6
model fields for unseen parameter combinations (see section 4.3). Note also that the ESEm CNN was designed for 2-D fields
and does not work as well for global averages.

### 225 3.4.2 Gaussian process emulator

A Gaussian process (GP) regression is a non-parametric approach that finds a distribution over the possible functions $f(x)$ that
are consistent with the observed data. It begins with a prior distribution and updates the prior distribution as new data points are
observed, producing the posterior distribution over functions. The priors are called kernels, or covariance functions. There are
several different kernels that can be used, for example constant, linear, radial basis function (RBF), expressing different prior
beliefs over the functional form of the model response. The kernel length-scale and the smoothness parameters (sometimes
refereed to as hyper-parameters) can then be fit using standard optimisation tools.

A key benefit of Gaussian process emulators over other approaches is that they can provide well calibrated uncertainty
quantification on their predictions. This is particularly important if the emulator is to be used for model calibration.

### 3.4.3 Random forest emulator

Random forest (RF) emulators generate a multitude of decision trees at the training time. The RF emulator creates several
decision trees by randomly picking samples to make decisions over, reducing the risk of overfitting. A feature of this approach
is that any predictions made must fall within the distribution of the training data by construction. That is, a RF regression model
cannot extrapolate beyond the training data.

## 4 Results

### 240 4.1 Spread across the PPE

First, we will illustrate the basic spread across the PPE for several key features of the simulated climate system. We first show
results for a few different outputs from the three scenarios. Figure 3 shows the ensemble zonal annual mean and $\pm 1$ standard
deviation ($\sigma$, shaded region) across the ensemble for aerosol optical depth (AOD: Figure 3A), column ice water path (IWP:





**Figure 4.** PDFs of global mean quantities from the simulations. A) aerosol optical depth (AOD), B) liquid water path (LWP), C) total cloud cover (CLDTOT), d) clear sky top of atmosphere net shortwave flux (FSNTC), E) shortwave cloud radiative effect (SWCF), F) longwave cloud radiative effect (LWCF), G) average cloud top number concentrtation (ACTNL) and H) top of atmosphere (TOA) flux residual. The PD - PI difference for aerosol forcing in is blue and SST4K - PD difference for feedbacks is in orange. Vertical dashed lines are the values using the default parameter set.





**Table 2.** Global averaged emulator statistics compared to the test data. Statistics shown are coefficient of determination ($R^2$) and root mean square error (RMSE).

| Emulator | LWCF $R^2$ | LWCF RMSE ($Wm^{-2}$) | SWCF $R^2$ | SWCF RMSE ($Wm^{-2}$) | RESTOM $R^2$ | RESTOM RMSE ($Wm^{-2}$) | LWP $R^2$ | LWP RMSE ($kg\,m^{-2}$) |
|---|---|---|---|---|---|---|---|---|
| NN | 0.72 | 3.08 | 0.74 | 5.52 | 0.79 | 4.14 | 0.73 | 0.019 |
| GP | 0.82 | 2.59 | 0.76 | 5.21 | 0.82 | 3.82 | 0.90 | 0.019 |
| RF | 0.57 | 3.71 | 0.54 | 7.31 | 0.57 | 5.85 | 0.78 | 0.027 |
| CNN | 0.70 | 3.46 | 0.73 | 5.62 | 0.80 | 4.17 | 0.69 | 0.033 |

Figure 3B), vertically integrated accumulation mode sulfate mass (BURDEN SO4: Figure 3C), vertically integrated cloud
condensation nuclei at 0.1% supersaturation (CCN 0.1%: Figure 3D), column liquid water path (LWP: Figure 3E), longwave
cloud forcing (LWCF: Figure 3F), cloud top number concentration for liquid (ACTNL: Figure 3G), and a histogram of the
global, annual mean net top of atmosphere flux balance (TOA: Figure 3H).

The zonal mean plots and histogram include all 263 members across each of the 3 run types (PD, SST4K and PI). Note that
the default case (dotted line) need not be near the ensemble mean (solid line), though it is generally within $\pm 1$ $\sigma$ of the mean.
This is not unexpected as the default parameter settings are not necessarily near the center of the range (see Table 1). Several
features stand out. First, the spread of IWP (Figure 3B) and LWP (Figure 3E) is large – roughly a factor of 2-4. Second, the
reduced SO4 burden, CCN and cloud top number (Figures 3C, 3D and 3G respectively) in the northern hemisphere in the PI
ensemble is clear. Also note that there is quite a spread in net TOA flux (Figure 3H). This means a large heat gain (positive) or
loss (negative) from the system. For PI and PD, most values are positive, while they are less positive for the SST4K ensemble.
A stable climate is possible in these configurations with large net TOA flux because there is an unbounded source/sink of heat
associated with the fixed ocean temperature, which constitutes $\sim$70% of the surface.

## 4.2   Forcing and feedback

One of the unique aspects of the PPE is that in addition to the control (PD) climate, since we run the same parameter sets with
perturbed climate, we can look at the variability of modeled climate responses. The differences from the PD to PI simulations
are only due to aerosol emissions (greenhouse gasses and SSTs remain the same). This enables us to look at the effects of
anthropogenic aerosols on climate. Aerosol effects comprise both direct scattering and absorption of radiation, as well as
indirect changes due to changes in cloud drop number from increased nucleation sites (Twomey, 1977) and further cloud
adjustments (Albrecht, 1989; Bellouin et al., 2020). Simulations with +4K uniformly warmer SSTs (SST4K) are commonly
used to look at fast feedbacks in the atmosphere in response to surface warming (Cess et al., 1989) and have been shown to be
generally similar to feedbacks with a more complete model treatment such as with a mixed layer ocean model (e.g., Gettelman
et al., 2012). To evaluate the forcing and feedback we use the weighted global mean of each ensemble member and subtract
the different run types (PD-PI and SST4K-PD, Figure 4).



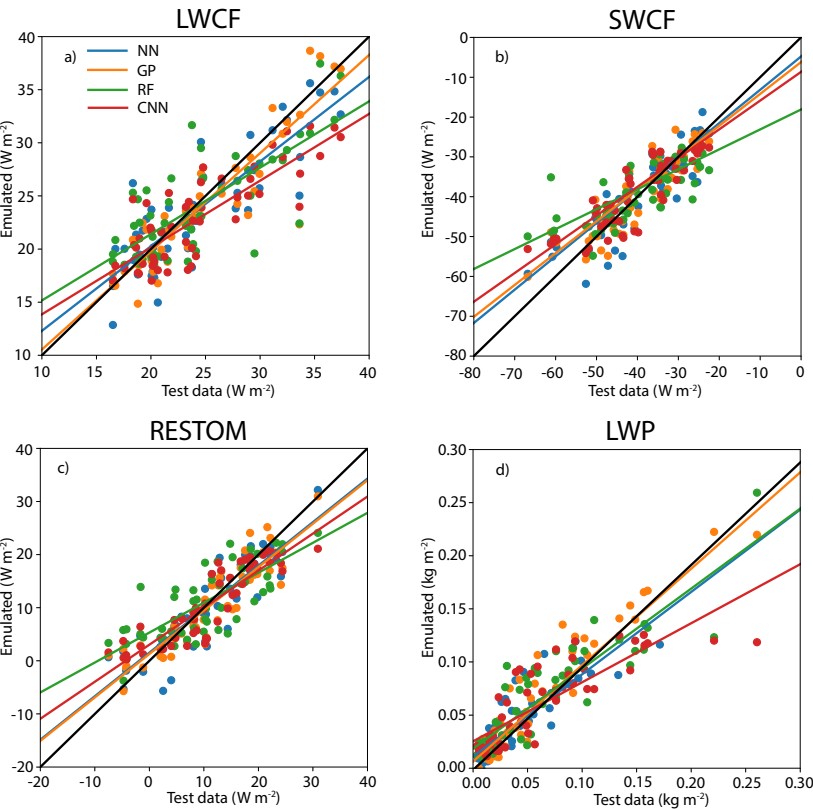

**Figure 5.** Comparison of global average emulator results against the global average test data. Lines are linear regression lines, except for the black line, which is the one to one line. Blue is the Columbia NN, orange is GP, green is RF and red is CNN. The root mean square error and coefficient of determination related to these results are shown in Table 2

.

Focusing on the aerosol forcing (PD - PI, blue), the difference in clear sky TOA shortwave radiation (FSNTC, Figure 4D) is a measure of the direct effect of aerosols and is about -0.4 Wm$^{-2}$. There is an increase in aerosol optical depth (AOD) (Figure 4A)
without much spread among the different parameter samples, and an increase in LWP (Figure 4B), with some parameter sets producing very small increases but with a longer tail of the distribution. The ensemble average TOA flux change (Figure 4H) is similar to the default parameter set at about -1.5 Wm$^{-2}$, but some sets have aerosol forcing of lower magnitude than -1 Wm$^{-2}$ and some more than -2 Wm$^{-2}$. Given the large diversity in model state (e.g. factor of 2-4 difference in LWP and IWP, and TOA differences up to 40 Wm$^{-2}$ (Figures 3B, 3E and Figure 3G respectively), it is remarkable that the histogram of TOA net forcing
is nearly Gaussian around the default value and with a range of only -2.5 to 0 Wm$^{-2}$. This is close to the assessed range of aerosol forcing by Bellouin et al. (2020), although we do not explore uncertainty in absorbing aerosol (such as black carbon) which would be expected to increase the tail of uncertainty to encompass positive forcing values. Other fields are similarly





distributed for PD-PI with the exception of the change in cloud drop number (Figure 4G), which drives cloud brightening and results in cloud adjustments. Also note that changes (both large and small) in cloud radiative effects in the SWCF and LWCF

are nearly opposite to each other. This may be due to high cloud changes: high clouds have large SW and LW effects which are opposite, so larger LW changes would be offset by SW changes. It is still noteworthy that there is not more spread. Also, it is interesting that the cloud top number change PD-PI has a significant spread (Figure 4G).

For feedback (SST4K - PD) results (orange in Figure 4), most of the distributions are slightly broader compared to the aerosol forcing. The larger magnitude of TOA difference in SST4K - PD (Figure 4D) is likely due to the large extra heat source

of emission from the warmer ocean. This is consistent with the absolute magnitude of changes in LWCF being larger in SST4K - PD than PD - PI (Figure 4F), while the absolute magnitude for the change in SWCF is similar (Figure 4E). There is generally a decrease in cloud fraction and an increase in outgoing clear sky LW radiation. There is a positive change in SWCF (which has a negative magnitude, Figure 4E) and a negative change in LWCF (which has a positive magnitude, Figure 4F), representing a weakening of cloud forcing consistent with loss of clouds. In the CESM2-CAM6 PPE, every simulation loses clouds with

the 4 K increase in SSTs (Figure 4C), and almost all have the same sign of cloud changes. This is a representation of positive cloud feedbacks seen in CESM2 and other models (e.g., Zelinka et al., 2020).

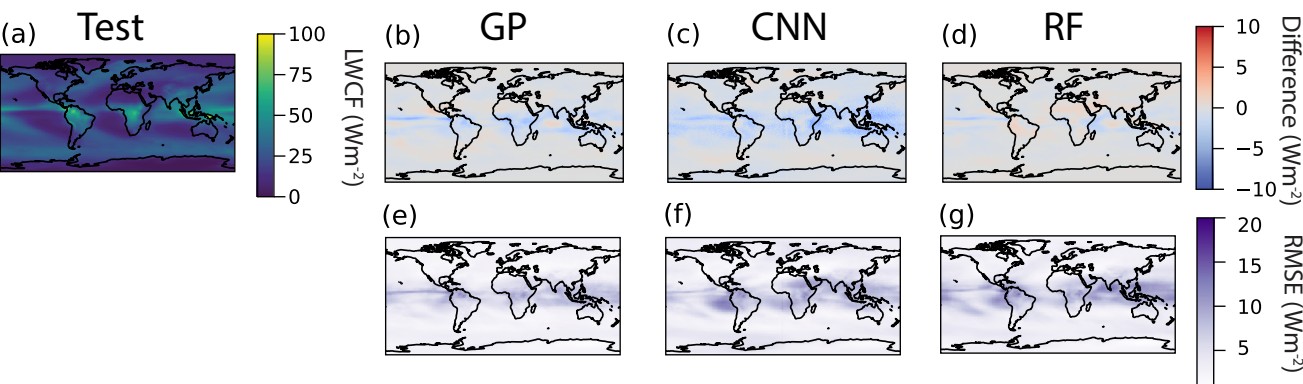

**Figure 6.** Emulated two-dimensional LWCF outputs using ESEm. (a) shows the mean of the test data (52 simulations), and (b), (c) and (d) are the difference between the test data and the emulated results for GP, CNN and RF, respectively. (e), (f) and (g) are the RMSE of the emulators using the same parameter sets as the test data for GP, CNN and RF, respectively.

### 4.3 Emulator results

Running climate models for large numbers of simulations can be computationally expensive. With the wealth of information from our PPE experiment, we can instead use emulators trained on the PPE data to obtain more insight into how the model

behaves, how to optimize it, and address scientific climate questions. In section 3.4, we described the different emulators used



**Table 3.** 2-D ESEm emulator statistical results compared to test data. The statistic shown is the root mean square error (RMSE).

| Emulator | LWCF 2-D RMSE (W m$^{-2}$) | SWCF 2-D RMSE (W m$^{-2}$) | RESTOM 2-D RMSE (W m$^{-2}$) | LWP 2-D RMSE (kg m$^{-2}$) |
|---|---|---|---|---|
| GP | 4.35 | 8.12 | 6.14 | 0.007 |
| RF | 5.54 | 9.34 | 8.21 | 0.008 |
| CNN | 5.39 | 11.23 | 6.91 | 0.009 |

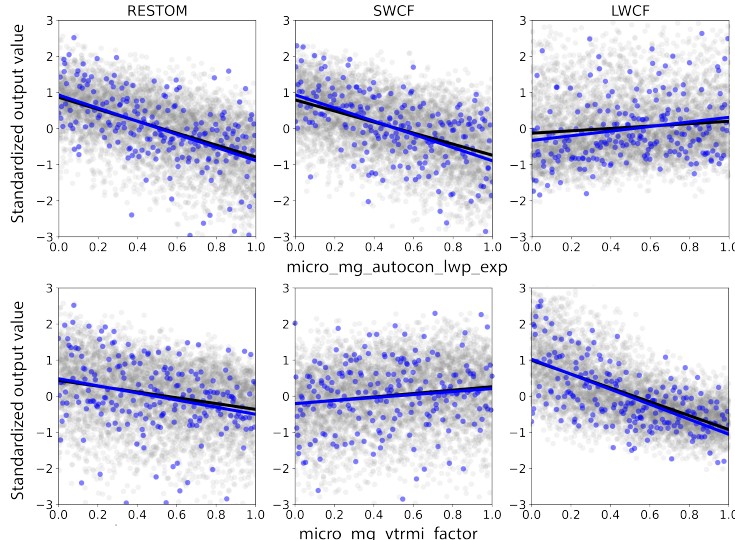

**Figure 7.** Example of output dependence on parameter values. Top: autoconversion of cloud droplets to rain. Bottom: fall speed for ice. Blue lines and dots represent the full PPE ensemble and black lines and gray dots are the emulated results using the Columbia NN emulator. Outputs are standardized and parameter values are normalized.

here. We focus on four different outputs when evaluating the emulator results: LWCF, SWCF, the residual top of model energy balance (RESTOM) and LWP. We emulate the response of model output to perturbations of all parameters in Table 1. Three of the emulators are from the ESEm package: Convolutional Neural Network (CNN), Gaussian Process (GP) and Random Forest (RF). The fourth emulator is the Columbia neural network (NN). The NN emulator was trained on 16 outputs simultaneously

while the CNN, GP and RF emulators were trained on each individual output separately. Figure 5 shows the global mean of the emulated results against the 52 PPE test ensembles, while Table 2 shows the error statistics (coefficient of determination ($R^2$) and root mean square error (RMSE). Note that for this example, the Columbia NN and GP are emulated with global mean values, while the RF and CNN are emulated over the two dimensional field, where the global mean is calculated after emulation. Recall that the CNN emulator is built for emulating 2-D fields, and cannot be used to emulate over global means.

For most of the outputs, the ESEm GP and Columbia NN emulators provide the best results. They have the highest $R^2$ values





and the lowest RMSE values (Table 1). The CNN emulator also has high $R^2$ values, however, the RMSE values are slightly higher compared to the Columbia NN and GP emulators. The RF emulator gives the lowest score.

As stated, the ESEm tool is also able to emulate 2-D fields. Figure 6 shows an example of the 2-D results with the ESEm emulators for LWCF. Figure 6a shows the mean of the 52 test simulations, while Figures 6b-d shows the difference between the
emulated results and the test simulations. Figure 6e-f shows the RMSE. The total average RMSE of LWCF along with SWCF, RESTOM and LWP are also shown in Table 3. In these cases, as when considering the global average, the GP emulator have the lowest RMSE. In this case (as opposed to GP emulation of global means in Figure 5), the GP is emulated over the 2-D fields. However, again, we find that the GP has the best performance compared to the RF and CNN.

### 4.4   Sensitivity of present day climate (PD) to parameters

With the large PPE, we can evaluate which parameters have the most impact on various outputs. In the following discussions we will show results from the full PPE and the Columbia NN emulator. Along with the ESEm GP emulator, the Columbia NN emulator typically had the lowest RMSE for the global averaged outputs when compared with the test data as shown in Table 2. Figure 7 shows how LWCF, SWCF and RESTOM depend on values of the cloud to rain autoconversion exponent parameter ($micro\_mg\_autocon\_lwp\_exp$) and the scaling parameter for fallspeed of cloud ice and snow ($micro\_mg\_vtrmi\_factor$).
The parameter values (x-axis) are normalized (scaled by the minimum and maximum parameter values) while the output values (y-axis) are standardized (scaled by the mean and standard deviation of the output values). The blue colors represent the entire PPE (263 samples) and the black/gray colors represent Columbia NN emulated results, using 5000 parameter sets. By evaluating the linear regression slope of the standardized outputs, the parameters with the largest slopes (in absolute terms) are determined to have the largest impact on the outputs. Since the outputs are standardized and parameters are normalized, the
slopes for different outputs and parameters are directly comparable. Slopes can be calculated for all outputs and parameters. For the example in Figure 7, it is clear that the cloud ice particle fallspeed parameter has a large impact on LWCF, while the autoconversion parameter value is important for RESTOM and SWCF. In the cases shown here, the regression slope from the PPE ensemble and the regression slope from the emulated results are almost identical. This indicates that the emulator can reproduce the spread of the PPE well. We note that for most parameter and output sets we produced, the emulator well
reproduced the PPE regression slope (not shown). Note that with this simple evaluation we can obtain the regression slope directly from the PPE. Thus, a full emulation to obtain the regression slope is not necessary, and the emulation results are included here primarily to illustrate performance of the emulator. We also acknowledge that the assumption that the outputs change linearly with the parameters is not necessary true in all instances; however, this assumption is reasonable for an initial evaluation. Furthermore, when looking at the coefficient of determination, it is evident that higher values are correlated with
steeper slopes (not shown).

Figure 8 shows a grid plot of the linear regression slopes (lines in Figure 7) for 16 outputs (vertical axis) against the 43 parameter values (horizontal axis). Blue values mean that the output decreases with increasing parameter value and red means that the output increases with parameter value. The darker the colors are, the steeper the slope is and the more the output is dependent on the parameter value. Since weather and climate systems can be different in different regions, we show global





**Figure 8.** Normalized linear regression slope for 16 outputs (y axis) against all parameter values (x axis). The global mean results as well as four different regions are shown; Arctic, Midlatitudes, Tropics and the Southern Ocean. The parameters are grouped into deep convection, aerosol, microphysics and turbulence parameters.



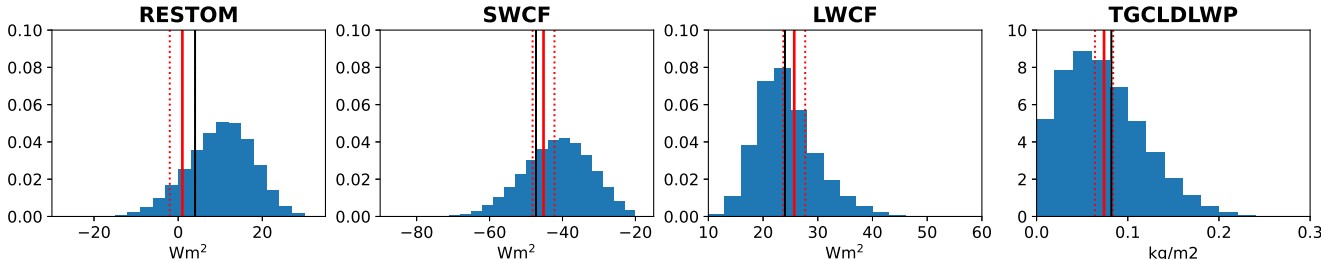

**Figure 9.** Histograms of outputs emulated using the Columbia NN. Red solid lines are the global means from the Clouds and the Earth's Radiant Energy System (CERES) Multisensor Advanced Climatology of Liquid Water Path (MAC-LWP). Dashed lines are the target range for tuning and the black lines are the values in the default simulation.

results as well as results from the Arctic, Midlatitudes, Tropics and the Southern Ocean. The black vertical lines divide the parameters into their respective physics package; deep convection, aerosol, microphysics and turbulence. The parameters are listed in the same order as in Table 1. The outputs (vertical axis going down) are listed in order of radiation, cloud properties and aerosol properties.

Some parameters stand out in almost all regions for many of the outputs, especially the microphysical parameters. The ac-
cretion enhancement factor ($micro\_mg\_accre\_enhan\_fact$), the autoconversion scaling factor ($micro\_mg\_autocon\_fact$), and the autoconversion exponent ($micro\_mg\_autocon\_lwp\_exp$) all directly affect rain formation and the amount of liquid water in the atmosphere. The autoconversion size threshold of cloud ice to snow ($micro\_mg\_dcs$) and the ice sedimentation factor ($micro\_mg\_vtrmi\_factor$) strongly influence the ice water path. However, in the tropics, where the deep convection scheme has a dominant influence, only the $micro\_mg\_vtrmi\_factor$ remains as an important microphysical parameter for
most of the outputs. Nonetheless, the parameter $micro\_mg\_dcs$ is still important for the LWCF in the tropics. This is not surprising since it has a large impact on cirrus clouds.

The deep convection parameter most impacting radiation outputs is the convective parcel temperature perturbation ($zmconv\_tiedke\_add$), and this is especially true in the Tropics. The triggering threshold for convection ($zmconv\_capelmt$) affects the ice water path and the sulfate burden. These may be related through sulfate effects on homogeneous nucleation of ice. Aerosol parameters
have less impact on radiation outputs, but several of them are important for cloud properties and precipitation. The turbulence parameters have a relatively lesser impact on the outputs presented here compared to the microphysics, aerosol and convection parameters. This might be because the selected range for the some of the key CLUBB parameters (like $clubb\_gamma\_coef$) are narrower compared to those used by others. For example, other PPE approaches with different versions of CAM have found the shallow cloud turbulence to be important (Guo et al., 2015) with a broader range of some CLUBB parameters.





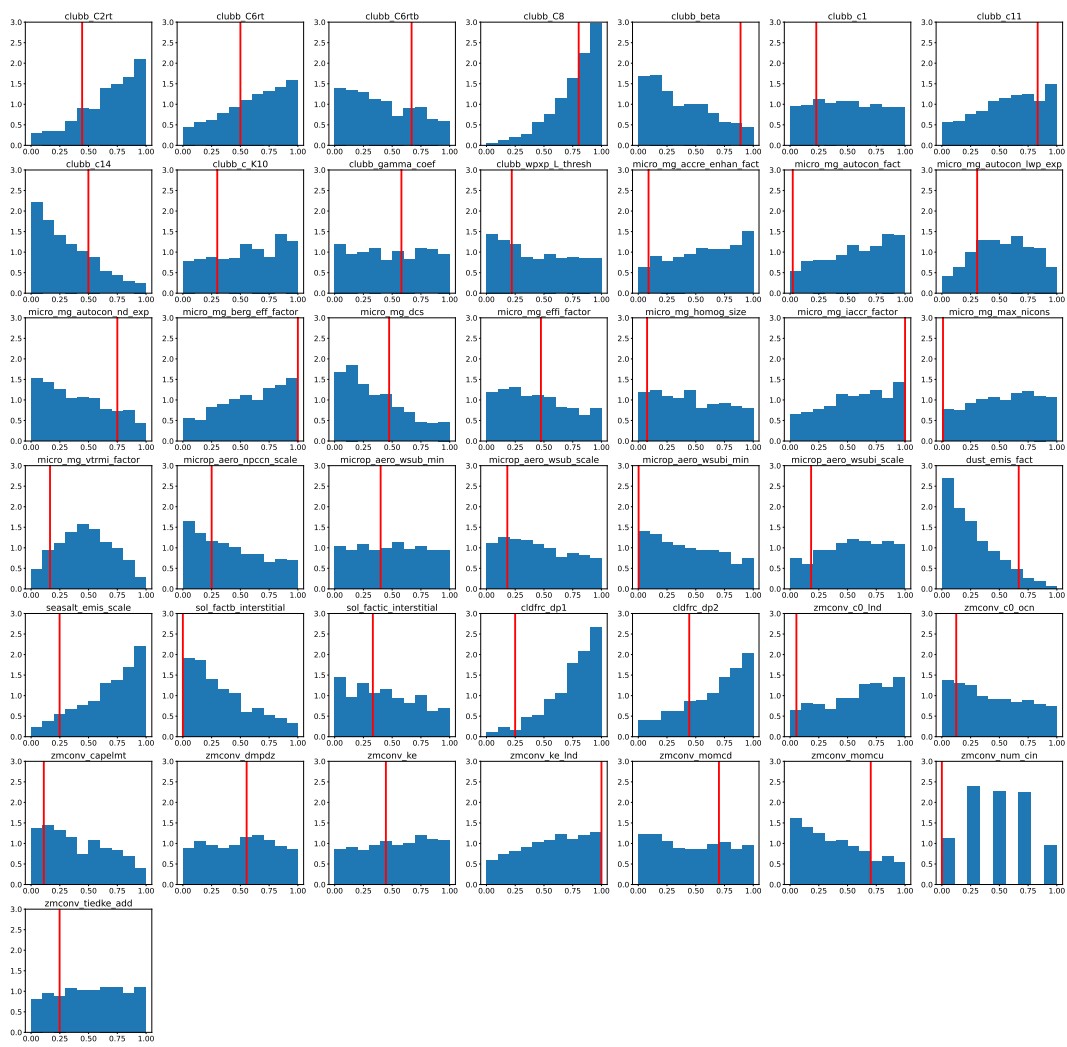

**Figure 10.** Histograms of parameter values producing outputs that fall within the desired targeted range surrounding the observations (as shown in Figure 9). The red solid lines are the default parameter values. Note that the values for $zconv\_num\_cin$ are integers and to five values, therefore the histogram is not continuous.



## 4.5 Tuning Example

One of the goals of PPE studies is to assist with constraining ('tuning') parameters in models. Though this is not the main goal of this paper, we experimented with tuning the CESM2 CAM model against the CERES and Multisensor Advanced Climatology of Liquid Water Path (MAC-LWP; Elsaesser et al., 2017) products using the Columbia NN emulator. To obtain enough samples we used 20,000,000 parameter samples with the emulator, creating the parameter samples using Latin hypercube sampling technique with all parameters normally distributed. For tuning, we focus on LWCF, SWCF, RESTOM and LWP and Figure 9 shows distributions of the emulated outputs. The targets are the observed global means (RESTOM: 0 Wm$^{-2}$, SWCF: -45.2 Wm$^{-2}$, LWCF: 25.7 Wm$^{-2}$ and LWP: 0.065 kgm$^{-2}$), indicated by the red solid lines in Figure 9. We look for all emulated outputs that are within the CERES mean $\pm$ 2 Wm$^{-2}$ for LWCF, $\pm$ 3 Wm$^{-2}$ SWCF and RESTOM and $\pm$ 0.01 kgm$^{-2}$ for LWP. The ranges are chosen to allow for enough samples to fall within the ranges in order to produce meaningful PDFs in Figure 10, while these values could be set to correspond to observational or emulator uncertainties. All parameter sets that are within the range for all four outputs are accepted; this effectively defines a bounded uniform likelihood over the 4-dimensional observational space. Figure 10 shows the histograms of the parameter values that results in outputs within the selected ranges. The red solid lines in Figure 10 indicate the default parameter value in CESM2-CAM6.

For parameter histograms that are non-uniform and strongly peaked, relatively more samples near the peak are within the acceptable tuning range. For example, the $clubb\_C8$ parameter peaks at relatively large values while $clubb\_c14$ parameter peaks at small values. But again, this result might be due to the relatively narrower chosen range for clubb parameters compared to the other types of parameters. Also interesting is the fact that a relatively low dust emission factor but high seasalt emission factor most often produce outputs consistent with the observations within the acceptable range. We emphasize that parameters with a peaked histogram in Figure 10 are *not* necessarily the parameters for which the outputs are most sensitive as determined by the regression slope magnitudes (outputs regressed on parameter values) in Figure 8. For instance, the dust and seasalt emission factors have strongly peaked histograms giving outputs in the acceptable tuning range, but relatively small regression slopes (Figure 8). This seemingly discrepancy can be explained by the linear nature of regression versus the nonlinear emulator. There is evidence that there is a complicated relationship between different parameters.

Ice fall speed ($micro\_mg\_vtrmi\_factor$) and the number of levels of convective inhibition in the deep convection scheme ($zmconv\_num\_cin$) are the only parameters with a strong peak in the middle of their range. As previously stated, the ranges were chosen by expert elicitation with the default values within the range minimum and maximum. However, the values giving realistic outputs here are most often near the edge of the physically plausible parameter ranges as determined by expert guidance. Some of the parameters, such as $clubb\_C8$ and $clubb\_C11$, have default settings near the upper end of the ranges which are close to the histogram peaks. Other parameters, such as $clubb\_c14$ and $dust\_emis\_fact$, have default settings near the middle of the range but strongly peaked histograms at the edge. Thus, these parameters most often have values at the edge of their range, different from the default values, to produce outputs consistent with observations. One possible explanation for this behavior could be that there are structural errors in the model and thus we must push some parameter values to the edge of their plausible range to obtain results consistent with observations. On the other hand, several parameters are fairly





uniform over the entire parameter range, such as for example $clubb\_c1$ and $zmconv\_tiedke\_add$. These results indicate that
any value of these parameters within their range can produce outputs close to observations. It is possible that by considering
more observational targets, these parameters could be further constrained and unreasonable parameter combinations could be
eliminated.

## 5  Summary and Conclusion

Here we have presented a CESM2-CAM6 perturbed parameter ensemble (PPE). We perturbed 45 parameters in the micro-
physics, turbulence, deep convection and aerosol physics packages and generated an ensemble with 263 members. Simulations
were generated for current climate, pre-industrial aerosol loading and future climate with 4K added to the sea surface temper-
ature. The main objective of this manuscript is to provide a description of the CESM-CAM6 PPE dataset and present some
initial results. The main results can be summarized as:

– The PPE has many different usages, for example, understanding uncertainties in model parameterizations, climate sen-
sitivities to parameter values, and optimal parameter tuning. The CESM2-CAM6 PPE data are publicly available for the
community to use. The CESM2-CAM6 PPE is extensible and new PPE data sets can be created in a straightforward way
using other parameter combinations or different model setups.

– Of the outputs evaluated here, there is a large spread in IWP, LWP, TOA among the individual ensembles (Figure 3).
Large TOA fluxes in many ensemble members are possible only because with fixed SSTs there is an unbounded heat
source/sink at the ocean surface that stabilizes the climate. Large ranges in LWP and IWP indicate that some parameters
can significantly increase or decrease cloud cover, although there is more constraint on the radiative fluxes since the
radiative forcing is non-linear with respect to cloud mass.

– Both aerosol forcing (PD-PI) and cloud feedback (SST4K-PD) show a spread in the output values considered here
(Figure 4). However, the aerosol forcing range is relative narrow compared to the cloud feedback (except for the cloud
top number concentration). There is more spread in the total cloud cover (CLDTOT) and LWP in the cloud feedback
case. This drives the top of atmosphere flux (TOA) differences as the cloud environment varies more with SST4K .

– We tested various emulators that were applied to the PPE ensemble. The Colombia Neural Network (NN), ESEm Gaus-
sian Process (GP) emulator, and ESEm Convoluted Neural Network (CNN) all produce reasonable results for selected
outputs, while the ESEm Random Forest (RF) emulator had the lowest scores when considering global means. Both the
CNN and RF outputs were emulated on 2-D fields, while the error statistics were calculated on the global mean values.
When calculating the error statistics on the 2-D fields, the RF performed at times better than the CNN emulator, while
the GP emulator still had the best score overall.

– With the large number of parameters, we evaluated the sensitivity of global outputs when changing the parameter values.
There were a select number of parameters that have strong sensitivity, especially several microphysics parameters. The



pattern changes slightly when considering specific zonal regions, such as the Arctic, Midlatitudes, Tropics and the
Southern Ocean. For example, the microphysics parameters create higher sensitivity in the Arcic and Midlatitudes than in
the Tropics and Southern Ocean, while some deep convection parameters have more impact in the Tropics and Southern
Ocean.

– We provided a simple tuning experiment using the Columbia NN emulator. We identified the parameter combinations
that gave results within a small range of observed global values and evaluated distributions of parameter values from
these combinations. A few parameter distributions peak within the range of physically plausible parameter values (as
determined by expert guidance), while several parameters peak at the edge of the parameter ranges. Only 4 observational
targets were used to constrain parameter values. By including more observations, parameters may be better constrained.
Furthermore, more elaborate techniques for sampling constrained parameter values, such as Markov chain Monte Carlo
and other Bayesian approaches, could improve the efficiency and accuracy of tuning, and allow for more comprehensive
account of observational uncertainties.

*Code and data availability.* The PPE dataset and the CESM2 CAM code version cam6_3_026 are available at the Climate Data Gateway
at NCAR (https://doi.org/10.26024/bzne-yf09 (Eidhammer et al., 2022)). The current version of CESM is available from https://github.com/
ESCOMP/CESM under the licence found here: https://www.cesm.ucar.edu/models/cesm2/copyright. The specific CERES data used in this
manuscript is available on Zenodo (https://doi.org/10.5281/zenodo.10426438 (Eidhammer and Gettelman, 2024)). The MAC-LWP data is
available at Goddard Earth Sciences Data and Information Services Center (10.5067/MEASURES/MACLWPM) (Elsaesser et al., 2017)

*Author contributions.* TE conducted the simulations to create the PPE, conducted analysis of the data and wrote the manuscript. AG initiated
the idea of the CESM PPE, oversaw the production of the PPE, conducted analysis of the and assisted in writing the manuscript. KT generated
the scripts to run the PPE simulations and assisted in editing the manuscript. DWP provided the ESEm emulators and provided input on the
manuscript. GE and MLW provided the Columbia Emulator. HM initated the idea of the CESM PPE, provided input on the work, and edited
the manuscript. DM and CS provided computing resources and input on the manuscript.

*Competing interests.* I declare that no competing interests are present

*Acknowledgements.* This research has been supported by the National Aeronautics and Space Administration (grant no. 80NSSC17K0073)
and the NSF STC Learning the Earth with Artificial Intelligence and Physics (LEAP), NSF Award Number 2019625. This material is
based upon work supported by the National Center for Atmospheric Research, a major facility sponsored by the National Science Foun-
dation and managed by the University Corporation for Atmospheric Research National Center for Atmospheric under Cooperative Agree-
ment No. 1852977. Computing resources were provided by the Computational and Information Systems Laboratory (2019) (CISL) on the





high-performance computing system Cheyenne (doi:10.5065/D6RX99HX) including through the Wyoming-NCAR alliance large allocation WYOM0124. We would like to acknowledge the use of computational resources (doi:10.5065/D6RX99HX) at the NCAR-Wyoming Super-computing Center provided by the National Science Foundation and the State of Wyoming, and supported by NCAR's Computational and Information Systems Laboratory.



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
