# Peer review of "An Extensible Perturbed Parameter Ensemble (PPE) for the Community Atmosphere Model Version 6"

_EGUsphere, 2023_

## Referee Comment (RC1)

**Review**

*An extensible perturbed parameter ensemble (PPE) for Community Atmosphere Model version 6.* Trude Eidhammer and Co-Authors.

This paper presents a clear description of the generation of a perturbed parameter ensemble (PPE) for a widely used global atmospheric model. Model sensitivity to key parameters is described. The use of the PPE to develop emulators characterizing the model's parametric dependencies is presented, and the emulators are applied to tuning. The paper is important in establishing new methods for model analysis and development, as well for its characterization of often poorly documented parametric dependencies. Only minor revisions are suggested.

**Minor Revisions**

1. The tuning results summarized in Figs. 9 and 10 point towards the possibility of using objective methods for model tuning going forward. Could the authors comment briefly on the possibilities of tuning towards a weighted function of a range of outputs, as would typically arise subjectively during a model development process? Have any experiments along these lines been undertaken?

2. In Eq. (1), as the Euclidean distance $d$ characterizes each ensemble member $m$, would it be useful to denote as $d_m$ rather than simply $d$?

3. ll. 139-142: Instead of as stated in the text, does *clubb_C6thl* refer to liquid water potential temperature flux, while clubb_C6rt is total water flux?

4. Longwave and shortwave cloud forcing are now more typically referred to as longwave and shortwave cloud radiative effective effect, to avoid confusion with forcing as a concept related to change in atmospheric composition.

5. Fig. 3H: I was looking for 3 histograms, one each for PD, SST4K, and PI. But there seem to be more. Perhaps this is just a perception issue related to overlapping colors. Perhaps better to outline the histograms rather than shade for clarity?

6. l. 359: Guo et al. (2015) used a GFDL model, not CAM, which could also explain the differing responses to CLUBB tuning. See also the discussion of CLUBB tuning for CAM in Appendix A of Bogenschutz et al. (2013).

l. 121: "aerosol" -> "aerosol effective"
l. 123: "3 and 5" -> "3 and 5 yrs"
l. 413: "relative" -> "relatively"

---

## Author Comment (AC2)

*We would like to thank the reviewer for positive and helpful comments and suggestions. Our responses and actions are indicated in blue below:*

This is an easy-to-read and useful introduction to NCAR's perturbed parameter ensemble (PPE) methodology. The results are interesting and important. However, I think that some additional discussion by the authors could clarify the interpretation of the results.

Major comments:

Lines 77–78: "we initially created 250 different sets of parameter values in addition to the default CESM2-CAM6 setup (total of 251 sets)." How was this number chosen? What is the consequence if only half of the 250 parameters sets are emulated? The paper contains a nice comparison of emulator techniques, but I wonder if an excellent emulator could be thwarted by a sparse sample of parameter sets.

*The number of ensembles were chosen based upon a parameter to ensemble numbers factor of approximately 5. This follows a study by Regayre et al. (2023), which used a ratio of about 6.*

*Regayre, L. A., Deaconu, L., Grosvenor, D. P., Sexton, D. M. H., Symonds, C., Langton, T., Watson-Paris, D., Mulcahy, J. P., Pringle, K. J., Richardson, M., Johnson, J. S., Rostron, J. W., Gordon, H., Lister, G., Stier, P., and Carslaw, K. S.: Identifying climate model structural inconsistencies allows for tight constraint of aerosol radiative forcing, Atmos. Chem. Phys., 23, 8749–8768, https://doi.org/10.5194/acp-23-8749-2023, 2023.*

*We added this text:*
*The ratio of number of ensembles to numbers of parameters is ~5.5, close to the ratio of 6 used in Regayre et al. 2023.*

*The comment regarding the consequence of numbers of ensemble members is an important one. Ongoing work indicates that the number might matter in that it is possible that parts of the parameter space are missed with sparse sampling. It seems that some types of emulator techniques perform better than others when sampling is sparse. Indeed there are many questions related to the construction of a PPE whose goal is parameter estimation. Number of samples is one, another is sampling strategy (LHS vs. CliMA's Ensemble Kalman methods vs. PPE-plus-first-guess-CPE iterative approach). There is also the question of whether to emulate "raw" outputs or some distillation of them. Finally the effect of a probabilistic emulator—whether GP or an ensemble of for example NNs—is also of critical importance. Answering the question of effect of ensemble size should ideally be tackled in the context of addressing some, most, or all of these questions, but that is outside the scope of the current work. However, work is being done on comparing the CAM6 PPE with another PPE with more ensemble members and comparing how they do perform with different emulators.*

Line 241: "First, we will illustrate the basic spread across the PPE for several key features of the simulated climate system." Is this the ensemble spread over all values based on a uniform sample of parameter values within the expert-chosen ranges?

*Yes. We have added the sentence:*
The ensembles are spread over all parameter values based upon the uniform sampling of the parameter values within the expert chosen ranges. The magnitude of the spread in output values is dependent both on the sensitivity of the parameter, the range of the parameter, and the combinations together with other parameters.

Could you provide some comments on how these spreads should be interpreted? The magnitude of the spread would be expected to depend sensitively on the chosen parameter range. If that range is chosen subjectively, then the spread will inherit that subjectivity. In principle, the range might be problematic because, e.g., we know that some ensemble members produce an unrealistic climate, because the authors state on lines 409–410 that "First, we will illustrate the basic spread across the PPE for several key features of the simulated climate system". Given this, are the spreads realistic?
E.g., the max ice fall speed factor is 25 times the min value, suggesting that we don't know the ice fall speed within an order of magnitude. Is this true? To cite another example, the accretion enhancing factor varies by a factor of 100. Is this degree of uncertainty realistic?

*The magnitude of the spread in output values is dependent both on the sensitivity of the parameter, the range of the parameter, and the combinations together with other parameters. For example, the ice fall speed parameter, which is sensitive in LWCF (Figure 7), can have the same value of LWCF at the extreme end of the parameter values, depending on the values of the other parameters. But it is correct that certain ensemble members can create unrealistic climate values due to the combination of the parameter values, and some studies remove these members in their analysis (for example Duffy et al. 2023).*

*However, regardless of whether the ranges of a given parameter are realistic, specification of some a priori range is usually done from the perspective of univariate parameter variation. Because it is typically unknown how parameter perturbations will interact, when performing simultaneous perturbations, it is advantageous to consider wider ranges of parameters, because these will more fully elucidate the model's ability to produce compensating errors. Compensating errors, in turn, help to indicate where independent information on individual parameters (observations, a priori theoretical or laboratory information) may be needed to independently constrain parameters and break the compensation of errors.*

*Finally, the ice fall speed range is perhaps large. However, knowing that there is a large uncertainty in the bulk ice particle habit the model tries to represent, the uncertainty inherent for this parameter is large.*

*We added this text in paragraph 3.3:*

*Some of the chosen ranges are large. However, regardless of whether the ranges of a given parameter are realistic, specification of some a priori range is usually done from the perspective of univariate parameter variation. Because it is typically unknown how parameter perturbations will interact, when performing simultaneous perturbations, it is advantageous to consider wider ranges of parameters. These will more fully elucidate the model's ability to produce compensating errors. Compensating errors, in turn, help to indicate where independent information on individual parameters (observations, a priori theoretical or laboratory information) may be needed to independently constrain parameters and break the compensation of errors.*

Minor comments:

Lines 44-46: "in the current study we perturb 45 different parameters, which would require a minimum of 3.5·10^13 (2^45) simulations using OAT if each parameter was tested with only two values in all combinations." How are you defining "OAT" here? 2^45 sample points would fill the entire 45-dimensional volume of parameter space, which would involve perturbing all parameters simultaneously, not one at a time. Perturbing each parameter individually would yield only 2*45 samples, no?

*One at a time means that for each new ensemble member, we change only one parameter. Let's say we have 3 parameters. First ensemble uses parameters a,b,c. Next ensemble uses a+1,b,c. Then the next ensemble uses a+1,b+1,c, then a+1,b+1,c+1, then a,b+1,c….. Each new ensemble member only changes one parameter. To fill all parameters possibilities, we here need 8 members (2^3).*

Table 1: The max value of DCS is listed as 1.0e-6. Should it be 1000e-6?
*Correct, this is fixed.*

Line 247: "global, annual mean net top of atmosphere flux balance (TOA: Figure 3H)." The definition of TOA is unclear to me. Does a net downward flux have positive TOA? Or negative TOA? Is TOA different than RESTOM (line 297)?

*The TOA is the net flux at the top of the atmosphere, and yes, net downward flux has a positive TOA. RESTOM, on the other hand, is the net flux at the top of the model level. In the model, the TOA fluxes adds an additional layer above the top of the model to provide a more appropriate comparison with satellite observations. We added the sentence:*
*Note, RESTOM is similar to TOA energy balance, but at the top of the model as opposed to top of the atmosphere.*

Line 414: Replace "relative" with "relatively".
*Done.*

Line 417: Replace "Colombia" with "Venezuela".
*We have replaced Colombia with Columbia.*

---

## Author Comment (AC3)

*We would like to thank the reviewer for positive and helpful comments and suggestions. Our responses and actions are indicated in blue below:*

Review

An extensible perturbed parameter ensemble (PPE) for Community Atmosphere Model version 6. Trude Eidhammer and Co-Authors.

This paper presents a clear description of the generation of a perturbed parameter ensemble (PPE) for a widely used global atmospheric model. Model sensitivity to key parameters is described. The use of the PPE to develop emulators characterizing the model's parametric dependencies is presented, and the emulators are applied to tuning. The paper is important in establishing new methods for model analysis and development, as well for its characterization of often poorly documented parametric dependencies. Only minor revisions are suggested.

**Minor Revisions**

1. The tuning results summarized in Figs. 9 and 10 point towards the possibility of using objective methods for model tuning going forward. Could the authors comment briefly on the possibilities of tuning towards a weighted function of a range of outputs, as would typically arise subjectively during a model development process? Have any experiments along these lines been undertaken?

*It is in essence what is being done in this paper with the rejection sampling. Each output has its own defined permissible range, and so each is "weighted" differently. This is also what is done in Watson-Parris et al (2021, GMD, https://doi.org/10.5194/gmd-14-7659-2021).*

2. In Eq. (1), as the Euclidean distance d characterizes each ensemble member m, would it be useful to denote as dm rather than simply d?

*Thank you for this suggestion. We have renamed d as $d_m$. We also realized that there was a mistake in the description of the equation:*

For each individual ensemble $j$, we calculate the average distance of each parameter $i$ in ensemble $j$ to parameter $i$ in the other ensembles. Then $d$ is the sum of all Euclidean distances in ensemble $j$ divided by number of parameters ($pa$) and ensembles ($en$):

*The sentence should instead be as follow:*

For each individual ensemble $m$, we calculate the average distance of each parameter $i$ in ensemble $m$ to parameter $i$ in the other ensembles, $j$. Then $d_m$ is the sum of all Euclidean distances in ensemble $m$ divided by number of parameters ($pa$) and ensembles ($en$):

3. ll. 139-142: Instead of as stated in the text, does clubb_C6thl refer to liquid water potential temperature flux, while clubb_C6rt is total water flux?

*You are correct. We have corrected the description, and the text is now as follows:*

The parameters clubb_C6rt (clubb_C6thl) and clubb_C6rtb (clubb_C6thlb) are the low and high skewness of Newtonian damping of the total water flux (potential temperature flux). Decreasing these parameters tends to boost fluxes, producing a more well mixed layer, with minor effects on cloud brightness. The low skewness especially impact stratocumulus while the high skewness especially impacts cumulus. Similar parameters were perturbed simultaneously so clubb_C6rt=clubb_C6thl and clubb_C6rtb=clubb_C6htlb.

4. Longwave and shortwave cloud forcing are now more typically referred to as longwave and shortwave cloud radiative effective effect, to avoid confusion with forcing as a concept related to change in atmospheric composition.

*We have changed all LWCF/SWCF to LWRE/SWRE*

5. Fig. 3H: I was looking for 3 histograms, one each for PD, SST4K, and PI. But there seem to be more. Perhaps this is just a perception issue related to overlapping colors. Perhaps better to outline the histograms rather than shade for clarity?

*We have changed the figure to illustrate the 3 histograms better by outlining them instead of using the shading as suggested. See new figure below*

[Figure]

6. l. 359: Guo et al. (2015) used a GFDL model, not CAM, which could also explain the differing responses to CLUBB tuning. See also the discussion of CLUBB tuning for CAM in Appendix A of Bogenschutz et al. (2013).

*The Guo et al (2015) paper we refer to is this one, which uses CAM:*

*Guo, Z., Wang, M., Qian, Y., Larson, V. E., Ghan, S., Ovchinnikov, M., A. Bogenschutz, P., Gettelman, A., and Zhou, T.: Parametric behaviors of CLUBB in simulations of low clouds in the Community Atmosphere Model (CAM), Journal of Advances in Modeling Earth Systems,*
*7, 1005–1025, https://doi.org/https://doi.org/10.1002/2014MS000405, 2015*

*Perhaps the reviewer is thinking about the Guo et al. (2014) paper, which use the GFDL model:*

*Guo, H., J. C. Golaz, L. J. Donner, P. Ginoux, and R. S. Hemler (2014), Multivariate probability density functions with dynamics in the GFDL atmospheric general circulation model: Global tests, J. Clim., 27(5), 2087–2108.*

*Since the Guo et al. (2015) paper uses the CAM5, we keep the text as is.*

l. 121: "aerosol" -> "aerosol effective" *We have rephrased the statement to:*

"By performing these three sets of simulations with the same parameter sets, not only can we evaluate the output spread by perturbing parameters, but we can also evaluate the cloud feedback (difference between PD and SST4) and aerosol forcing (difference between PD and PI). Here the aerosol forcing is the aerosol effective forcing after adjustments of atmospheric temperature and humidity."

l. 123: "3 and 5" -> "3 and 5 yrs" Corrected.
l. 413: "relative" -> "relatively" Corrected.

---

## Author Response (AR2)

Dear Topic Editor

We have addressed the suggested technical corrections. Responses are in blue below:

- The abbreviation "Fig." should be used when it appears in running text and should be followed by a number unless it comes at the beginning of a sentence, e.g.: "The results are depicted in Fig. 5. Figure 9 reveals that...". Please consider this at multiple places within the manuscript.
  All figures are in text are now addressed as Fig. , and if the beginning of the sentence, as Figure

- The abbreviation "Sect." should be used when it appears in running text and should be followed by a number unless it comes at the beginning of a sentence. Please consider this at multiple places within the manuscript.
  Done

- Please ensure that figure labels have units, e.g.: in Fig. 3 the degree symbol (°) is missing in conjunction with Latitude.
  Figure 3 now have the degree symbol in conjunction with Latitude

  4) Labels of figure panels must be included with brackets around letters being lower case (e.g. (a), (b), etc.). Please consider this, since currently you use a variety of ways to denote individual panels on various figures: (a), a), A)... Please also correct the text when referring to figure panels accordingly.
  All labels are now bracketed as (a), (b) etc

  5) A few additional comments regarding abbreviations:

  a) The abbreviation "RMSE" is first used and subsequently defined multiple times within the text. Please correct this and omit unnecessary definitions. I encountered a similar issue with "2-D" (first used and subsequently defined).
  Done

  b) Several abbreviations ("TOA", CESM2", "CAM6", "PPE", ...) are also introduced more than once within the main text - please omit unnecessary definitions. Abbreviations need to be defined in the abstract (as you properly do) and then again at the first instance in the rest of the text (but please not more than once in the main text).
  Done. Note, we kept the definitions in a few of the figure captions of Figs., 3, 4 and Table 2).

  c) I would advise the following Title: "An Extensible Perturbed Parameter Ensemble for the
  Community Atmosphere Model Version 6" - without introducing "(PPE)" in the title.

Done
* * *
- Additional notes:

    1. We added two more citations of newly published papers that also use the PPE datasets described in this paper:

        Song, C., McCoy, D. T., Eidhammer, T., Gettelman, A., McCoy, I. L., Watson-Parris, D., Wall, C. J., Elsaesser, G., and Wood, R.: Buffering of Aerosol-Cloud Adjustments by Coupling Between Radiative Susceptibility and Precipitation Efficiency, Geophysical Research Letters, 51, e2024GL108 663, https://doi.org/https://doi.org/10.1029/2024GL108663, e2024GL108663 2024GL108663, 2024.

        Gettelman, A., Eidhammer, T., Duffy, M. L., McCoy, D. T., Song, C., and Watson-Parris, D.: The Interaction Between Climate Forcing and Feedbacks, Journal of Geophysical Research: Atmospheres, 129, e2024JD040 857, https://doi.org/https://doi.org/10.1029/2024JD040857, e2024JD040857 2024JD040857, 2024.

    2. I noticed that Figure 4 and 8 that in the revised manuscript did not have the correct updated Long/short wave radiative effect abbreviations. These figures still had SWCF/LWCF instead of SWRE/LWRE. This is now corrected in the final version.